# Differentiated Uses of Social Networking Platforms among Young People in the Autonomous Region of Madrid

**María Cruz López-de-Ayala** [1,*] **, Antonio García-Jiménez** [1] **and Yolanda Pastor-Ruiz** [2]

1. Department of Communication Sciences and Sociology, Rey Juan Carlos University, Tulipán s/n, 28933 Madrid, Spain; antonio.garcia@urjc.es
2. Psychology Department, Rey Juan Carlos University, Tulipán s/n, 28933 Madrid, Spain; yolanda.pastor@urjc.es
* Correspondence: mariacruz.lopezdeayala@urjc.es

**Abstract:** Young people make intensive and varied use of social networks, which depends on individual differences and use motives. This study analyses the different ways in which young people use the main social media platforms and the role that gender, age and social class play in users' behaviour with a representative sample of young people aged 17–24 in the Community of Madrid (Spain) (N = 533). Women and the youngest people (17–19 years of age) used social networks more consistently. Instagram and Facebook were the most widely used social networks with greater diversity of use in the areas of sociability and entertainment. While women and the younger age group spent more time using Instagram, young people from the upper class and upper-middle class used Facebook more often in a variety of ways. YouTube was used less frequently, and it was used mainly for entertainment purposes. Snapchat played a small role in this segment of the population. We conclude that age and social class play an important role in defining the different ways in which young people in the Region of Madrid use social media, and such influences vary depending on the platform under study and the types of activities being carried out.

**Keywords:** online social networking; youth; types of uses; age; gender; social class

## 1. Introduction

*1.1. Socio-Demographic Features and Variables for Measuring Frequency and Use of Activities*

Analysis of the ways in which different social media platforms are used by a given sector of the population is not an easy task, as this subject of study is prone to frequent changes (Fietkiewicz et al. 2016). In the case of young people who use these platforms daily, the current trend at the international level seems to indicate greater use of Instagram, followed by other platforms such as Snapchat or Facebook, as pointed out by some researchers (Knight-McCord et al. 2016; Huang and Su 2018). In fact, Alhabash and Ma (2017) mention Instagram as the platform where young people spend more time daily, followed by Snapchat, Facebook, and Twitter. At the same time, during 2020 (Qustodio 2020) an increase in Snapchat and TikTok consumption has been observed. However, it should be noted that differences could be seen in the role of each platform, depending on the country under study, and even according to the year of the investigation (Villanti et al. 2017). Studies carried out by specialized research companies in Spain indicate that an average of one hour per day is spent using social networks, with the most intensive use occurring among men and young people (16–30 years of age). According to frequency of use, Facebook (85%) and Instagram (76%) stand out as the platforms used mainly for social relations and entertainment (IAB 2019), although they are losing some weight due to the growing diversification in the consumption of social media platforms and increased consumption on other platforms (IAB 2020).

It is possible to identify distinctive profiles of young users of social media. In this sense, Blank and Lutz (2017) find that the variables of age and socioeconomic status are the

most determining factors in differentiated use of most social network platforms. In the case of Facebook, factors such as age, gender, and whether one has completed higher education has an impact on variation of use (Blank and Lutz 2017). Women, younger people (Blank and Lutz 2017), and people with lower levels of education use Facebook more often (Blank and Lutz 2017; Correa 2016). However, educational levels seem to indicate some differences in the way in which this platform is used. Thus, when the platform is used for purely social purposes, socio-demographic factors make no difference in its use. In the opposite way, young people with higher education levels and skills use it more often for information as well as political and civic participation (Correa 2016).

In this regard, (early) age is also pointed to as a key factor for increased use of TikTok (Bucknell and Kottasz 2020). Age and income also have an impact on the use of Twitter. In the case of Instagram, there are no significant demographic features in predicting its use among the British population (Blank and Lutz 2017).

*1.2. Motivations, Differentiated Uses, and Gratification*

The way in which young people use social media can be influenced by the meanings attributed to each social network. Instagram is considered a space where young people show the best of their personalities and their physical features as well. Facebook allows users to display themselves to others in an acceptable way. Finally, Twitter is used for information, with a certain sense of informality, and Snapchat is a space for spontaneous and playful networking (Boczkowski et al. 2018).

Alhabash and Ma (2017) have observed that young people are making equal use of Snapchat, Instagram, Facebook, and Twitter to share information. On all four platforms, the two most important reasons for using these networks are entertainment and ease of use. It has also found that fewer motivations are involved in predicting the intensity of use of Facebook and Twitter, while more incentives come into play in forecasting the frequency of use of Instagram and Snapchat. Specifically, on Instagram, Fidan et al. (2021) highlight communication and interaction, knowledge acquisition, and "entertainment and sharing", as key factors guiding its use and consumption. Even the consumption of Instagram has been associated with an interest in being informed (Tarullo 2020).

From another point of view, Instagram and Snapchat are media perceived to be more specialized than Facebook and Twitter, which might have an impact on the diversity of motivations sought and obtained from their use. This motivational difference, in the opinion of Oh and Syn (2015), who analyzed Facebook, Twitter, Delicious, YouTube and Flickr, may be affected by the diverse content and variety of purposes for which users engage in these platforms. If you compare Facebook and Snapchat, college students use Snapchat more often. At the same time, these students join Facebook when they want to build up networks of contacts, while peer pressure and content appeal encourage them to use Snapchat. Stanley (2015) also found that women use Facebook and Snapchat to oversee the lives of family and friends, while men join Facebook to make contacts and meet new people.

In the case of YouTube (Khan 2017), it has been observed that the strongest predicting factor for posting a like or dislike of a video was entertainment, and commenting on a video or uploading it to the platform was strongly related to the desire for interaction. Meanwhile, Balakrishnan and Griffiths (2017) have stated that social gratification has a significant influence on the creation and viewing of YouTube content. As far as Facebook is concerned, Kaya and Bicen (2016) highlight the fact that this platform is mainly used for entertainment purposes, as well as for sharing news, photos, and songs. Dhir and Tsai (2017) have suggested the possibility that the gratification involved in the process as well as the content play no significant role in IFU (intensity of Facebook use) prediction. Malik et al. (2015) note that the main reason why young people share photos on Facebook is to disclose personal information. At the same time, these researchers find that another of the main advantages of this platform is its use as a common leisure activity.

Regarding Twitter, Alrajehi (2016) point out that young people use it to meet new people, follow local news, and discuss and exchange opinions. In the case of growing social media platforms such as TikTok, participation is driven by a desire to express oneself, interact with others and escape from daily pressures. The motivation to produce videos on TikTok, would derive from the self-expression and archiving needs (Omar and Dequan 2020).

Finally, Snapchat seems to be more oriented toward the personal sphere (Kofoed and Larsen 2016). On this platform, people expose themselves largely and strengthen the bonds of friendship through shared photos that are not as refined as those on Instagram are. Specifically, everything points to the fact that Instagram caters to young people's need for social interaction and the desire to elude themselves. In this regard, gender, educational background, and perception about the willingness to interact with Instagram are the key elements in understanding how this platform is used (Huang and Su 2018).

In Spain, some studies have predicted a massive migration from Facebook to Instagram among young people (Marcelino 2015), and have highlighted the association with entertainment in the case of Facebook (Igartua and Rodríguez-de-Dios 2016). Some studies also have addressed the differentiated uses of language on these platforms (Candale 2017), or its connection with identity (Sanchez and Pérez 2016). On the other hand, young people perceive social networks primarily as spaces for information and for sharing opinions and experiences with the community, although they also show caution when opinions and personal information are posted. Their assessment depends on gender, age and family social class, and also predicts types of uses, but with limited effects (López-de-Ayala et al. 2020).

The primary objective of this study is to learn more about the way in which young people in the Autonomous Region of Madrid use social networks. This region has just over 6.7 million inhabitants, distributed mainly between the capital of Spain, large (>100,000 inhabitants) and medium-sized municipalities (>20,000 inhabitants) and, to a lesser extent, in small rural towns. It makes up around 14% of the Spanish population. Its particular characteristics make it necessary to carry out an in-depth study of trends in the use of social media. We are especially interested in knowing what kinds of activities are being developed on the different platforms, and in discovering the socio-demographic variables that affect the development of different online behaviour patterns. The research questions raised in this study are the following:

- How much time do young people in the Region of Madrid spend on the activity of social networking?
- What are the main social networking platforms they use? What is their level of activity on each platform?
- How does the amount of time devoted to social networks affect their preferences in using different platforms?
- How do young people in the Region of Madrid create balance in their use of the major social networking platforms—Facebook, Twitter, Instagram, YouTube and Snapchat?
- What activities do they carry out on each of these platforms, and what is the profile of each platform according to the way each one is used?
- How do gender, age, and social class affect the use of social networks by young people from the Region of Madrid?

## 2. Materials and Methods

### 2.1. Participants

A representative sample of young people between 17 and 24 years of age in the Autonomous Region of Madrid was used to carry out this study. The Autonomous Community of Madrid is one of the seventeen autonomous regions that comprise the Spanish State, it has a population of 6,779,888 inhabitants (Instituto Nacional de Estadística 2020) and includes the capital of Spain, Madrid. A two-stage sampling was applied, stratified by clusters, and selection of the primary sampling units (municipalities and census sections) was performed in a proportionate, random way, and the last units (individuals), by random routes and gender/age quotas. The sample base was designed using data corresponding to

the population interpolated by age, with the reference date for each gender being 1 January 2019 for the Autonomous Region of Madrid. The sampling error was set at $\pm 4$ for overall data ($p = q = 50$; confidence level of 95%).

The sample consisted of 533 respondents (51.2% boys and 48.7% girls; 36.6% were 17-19 years of age, and 63.4% were 20–24 years old). As a very slight deviation was observed between the distribution of the sample by age and gender with regard to what was presented by the statistical universe of young people in the Autonomous Region of Madrid, weights were included in the sample to adjust the results to the population data and maintain its representativeness.

### 2.2. Variables and Instruments

The sociodemographic variables considered were gender, age (17–19 and 20–24 years), and social class. The latter was developed using two variables: occupation and educational level of the father, assuming that he was the one who contributed the most income, except in those cases in which the father was unemployed, was a pensioner, etc. In those cases, the mother was considered the highest financial contributor.

A questionnaire created by the authors was used to collect information on the use of social networks and platforms. With the aim of learning more about the ways in which different platforms were used, and given the possibility of changing from one social network to another, leaving profiles open that were no longer in use, the young people were asked about an array of activities they are able to carry out on these platforms. The items included were based on a review of the state of the art, previous research and focus groups. By using this data, we have been able to identify the platforms on which young people carry out their activities, how these activities were distributed by platform, and the degree of activity developed on each one. The questions were closed, and the respondents could choose between different options.

### 2.3. Procedure

The surveys were conducted between 17 June and 4 July 2019, after obtaining a favorable report from our institution's ethics committee. The survey was implemented through the CAPI system (computer-assisted personal interview), and the duration of the questionnaire was approximately 25 min. The interviews were conducted personally in the homes of the respondents by a market research firm. The explicit consent of the respondent was requested after explaining the type of data that would be collected in the survey and what it would be used for, as well as informing them of its anonymous nature.

### 2.4. Data Analysis

The data was analysed using the statistical program SPSS, version 26, and the level of statistical validity was for the value of $p < 0.05$. The Chi-square test ($\chi 2$) was used to analyse the relationships (dependence–independence) between nominal variables (or ordinal variables with scant values), and qualitative variables, in which percentage data are described. For explanatory variables with several response categories, once the relationship was verified through $\chi 2$, the option of testing by pairs of the equality of column proportions was selected, offered by the SPSS custom table option.

The non-parametric Mann–Whitney U test have been applied to compare differences between two independent samples, when the dependent variable is either ordinal or continuous, but not normally distributed. According to Tomczak and Tomczak (2014), we use the Rosenthal correlation to measure how big the difference is between two groups (effect size). The Rosenthal correlation is a generic one that simply divides the standardized test statistics, by the square root of the sample size (Rosenthal 1991). The interpretation various rules of thumb exist. The interpretation rule of thumb from Bartz (1999, p. 184) is: 0.00 < 0.20 = very low; 0.20 < 0.40 = low; 0.40 < 0.60 = moderate; 0.60 < 0.80 = strong; and 0.80 < 1.00 = very strong.

In the case of scale or continuous variables (with means), a Welch *t*-test was used for independent samples when the explanatory variable offered two partitions. When the explanatory variable offered several partitions (ordinal variable), a one-way ANOVA was performed, if the assumptions of normality and homogeneity of variances (homoscedasticity) were met. When these assumptions were not met, a Kruskal–Wallis non-parametric test and, then, a U-Mann–Whitney test (two partitions) were performed, and subsequently a post-hoc test using Dunn's test with Bonferroni correction was carried out to compare the results 'two at a time' regarding the categories of the independent variable.

To facilitate understanding of the results, statistically significant differences were displayed in bold in the tables. Reference was made to these differences in the text only in those cases when they were statistically significant.

## 3. Result

### 3.1. Use of Social Networks

The young people of Madrid have stated that they use social networks intensively, which is clearly shown by the fact that only eight of the 533 respondents have said they do not use social networks (1.5%), yet most (53%) have claimed they use these networks continuously, with higher percentages among women: 57.6% compared to 48.5% for men ($\chi2 = 4.48$, df = 1, $p = 0.034$). Compared to this majority, 30.6% have stated that they use them several times a day; 10.7% use them a short time each day; and the rest (around 5%) use them less frequently. A U-Mann–Whitney test indicated that the differences by sex are significant for the whole population, but low, U ($n_1 = 273$, $n_2 = 260$) = 39,360, $p < 0.016$; $r = 0.10$), with a slightly higher scores of connection among women (*Mdn* = 7) versus men (*Mdn* = 6).

Regarding the differences by age group, a U-Mann–Whitney test indicated that the differences are significant but low for the whole population, U ($n_1 = 195$, $n_2 = 338$) = 28,603, $p = 0.005$, $r = -0.12$, with a slightly higher scores of connection among the younger age group (*Mdn* = 7) versus older age group (*Mdn* = 6). The youngest (17–19 years of age) said they use more these platforms continuously on a daily basis (60.5% compared to 48.4% of young people 20–24 years of age) ($\chi2 = 7.49$, df = 1, $p = 0.006$).

Taking into account social class, young people at higher social levels said their use of social networks was highly sporadic, less than once a week. The Goodman–Kruskal gamma ($\gamma = 0.015$, $p = 0.819$) and independent-samples Kruskal–Wallis test showed no statistically significant differences, $\chi^2$ (4, N = 533) = 3.92, $p < 0.417$.

### 3.2. Use of Social Networks by Platform

Young people have not abandoned Facebook which continues to be one of the platforms on which most of the young people surveyed are active (77.5%), and where they carried out the highest number of activities (*M* = 8.5, *SD* = 5.18) of those users who carry out some form of activity of the twenty mentioned and mode of 5. The other platform that is the top leader in the highest amount of activity is Instagram (78.7% of users, *M* = 7.0, *SD* = 4.34, mode = 5). Thirdly, 74.3% of young people were active on YouTube, with an average number of activities that was lower (*M* =4.1, *SD* = 3.31, mode = 1) than that of Facebook and Instagram. Following: Twitter (39.2% of users, *M* = 4.4, *SD* = 5.18, mode = 5); Snapchat (16.4% of users, *M* = 2.3 activities, *SD* = 1.79, mode = 1); and others (81% of users, *M* = 7, *SD* = 4.33, mode = 2).

The most common tendency among young people was to be present on more than one platform. Along these same lines, of those who say they are active on at least one of the five platforms mentioned (Facebook, Twitter, Instagram, YouTube and Snapchat), only 9.9% are active on just one of them; 28% are active on two; 37.6% on three; and 24.5% are active on all of them. If we add the category of 'others', only 3.7% have activities on a single platform, with the most common situation being that young people participate in at least four platforms (36.1%), considering that the category of "others" can gather several different platforms, or refer to only one. In short, in relation to the most heavily used

platforms, it has been observed that 60% of all respondents are active on Facebook and Instagram simultaneously, and half are active on Facebook, Instagram and YouTube (51%).

Frequency of social network use is also related to the preference for different platforms. Those who said they continuously use social networks tend to be more active on Instagram and YouTube compared to those who did not engage in continuous use (60.1% used Instagram compared to 26.2 which didn't use it, $\chi^2 = 41.16$, df = 1, $p < 0.001$; and 56.4% used YouTube compared to 42.9 which didn't use it, $\chi^2 = 7.1$, df = 1, $p = 0.007$). In short, looking at it from another perspective, 89.4% of Instagram users were continuously connected to social networks compared to 79.6% on other platforms, 79% on YouTube, 77% on Facebook and 41.7% on Twitter.

Regarding the frequency of use of these activities, 31.8% of those who said they use Facebook carried out more than ten activities on this network; 8.4% on Twitter; 19% on Instagram; and 5.1% on YouTube. On the latter, the maximum number of activities performed was seventeen, while the maximum number of activities carried out on Snapchat was eight.

Women ($M = 6$, $SD = 4.5$) were more active on Instagram than men ($M = 5.02$, $SD = 5.1$, $t(528) = -2.32$, $p = 0.021$) and on Snapchat ($M_{women} = 0.5$, $SD = 1.17$; $M_{men} = 0.28$, $SD = 1.1$; $t(1, 521) = -2.22$, $p = 0.027$). The oldest group were also more active on Facebook ($M = 7.2$, $SD = 5.72$) than the youngest group ($M = 5.62$, $SD = 5.78$; $t(413) = -3.12$, $p = 0.002$. However, no significant differences were observed in the degree of activity that occurred on the rest of the social networking platforms, either by gender or age group ($p > 0.05$) (see Table 1).

**Table 1.** Use of social networks by platform among young people from 17 to 24 years of age in the Autonomous Region of Madrid, according to gender, age group and social class (in percentages).

|  | Men | Women | 17–19 | 20–24 | Upper Class | Upper Middle Class | Middle Class | Lower Middle Class | Lower Class |
|---|---|---|---|---|---|---|---|---|---|
| Facebook | 77.8 | 77.3 | **70.6** | **81.7** [3] | **84.8** | **87.2** | **75.5** | **73.2** | **64.5** [5] |
| Twitter | 40.3 | 38.1 | 41.1 | 38.1 | **46.9** | **51.6** | **35.1** | **37.1** | **26.0** [6] |
| Instagram | **73** | **84.7** [1] | **83.5** | **75.8** [4] | 73.8 | 73.0 | 82.7 | 76.5 | 82.2 |
| YouTube | 73.3 | 75.3 | 77.2 | 72.5 | 73.2 | 63.4 | 75.6 | 77.9 | 84.9 |
| Snapchat | **10.2** | **22.9** [2] | 19.9 | 14.3 | 19.9 | 14.4 | 17.2 | 15.3 | 11.5 |
| Others | 82.8 | 79.1 | 81.1 | 80.9 | 76.9 | 85.8 | 82.0 | 76.3 | 86.4 |

Statistically significant differences for $p < 0.05$ appear in bold. [1] $\chi^2 = 10.88$, df = 1, $p = 0.001$; [2] $\chi^2 = 15.88$, df = 1, $p < 0.001$; [3] $\chi^2 = 9.27$, df = 1, $p = 0.002$; [4] $\chi^2 = 4.36$, df = 1, $p = 0.037$; [5] $\chi^2 = 10.76$, df = 4, $p = 0.029$; [6] $\chi^2 = 11.36$, df = 4, $p = 0.023$.

The Goodman–Kruskal gamma showed that there was a significant weak positive association between social position and the tendency to participate in Facebook ($\gamma = 0.143$, $p < 0.005$). A Kruskal–Wallis test showed that social class had a modest significant effect on how different activities young people do on Facebook, $\chi^2$ (4, N = 533) = 15.85, $p < 0.01$, $E^2 = 0.044$. A post-hoc test using Dunn's test with Bonferroni correction showed the significant differences between: lower-middle classes ($M = 7.8$) and upper-middle classes ($M = 8.7$) ($p < 0.01$) and upper classes ($p < 0.001$) ($M = 10.9$); middle classes ($M = 7.8$) and upper-middle classes ($M = 8.7$) ($p < 0.005$) and upper classes ($M = 10.9$) ($p < 0.001$). Thus, upper class youth not only participated to a greater extent on Facebook or Twitter, they also maintained an average amount of activities that was higher on Facebook in comparison to those in the middle and lower-middle classes, and the lower class had less activity compared to the lower-middle, middle, and upper-middle classes.

### 3.3. Activities Carried Out on Social Network Platforms

Nearly all of the young people (99.3%) said they watched videos and music on social networks; those who watched videos or see photos of friends were 99%; those who talk to friends were 98%, while those who talk to family was 94.4%; and finally, the percentage of those who publish or post personal content on their profile were 95%. Moreover, 92% searched for entertaining content. In contrast, just over half indicated that they played

online (53%), published opinions on social or political issues (56%), or posted criticism or complaints on a public profile occasionally (57%) (see Table 2).

**Table 2.** Distribution by platform of the activities carried out by young people from 17 to 24 years of age in the Autonomous Region of Madrid (% of all users who carry out some activity (left column), and the % of that amount who carry out some activity on each social network).

| | % of All Users | Facebook | Twitter | Instagram | YouTube | Snapchat | Others |
|---|---|---|---|---|---|---|---|
| Publish/upload personal content to your profile | **94.7** | **75.4** | 21.5 | **67.5** | 7.8 | 5.8 | 13.8 |
| Watch videos/see photos of friends and family | **99.1** | **66.9** | 14.7 | **60.1** | 18.1 | 6.7 | 16.7 |
| Talk to friends | **98.2** | **57.4** | 14.6 | **56.5** | 5.8 | 4.9 | 31.5 |
| Talk to family members | **94.4** | **54.8** | 8.1 | 38.9 | 3.7 | 1.9 | 36.7 |
| Watch videos/music | **99.3** | 41.8 | 9.2 | 40.3 | **64.3** | 2.6 | 15.8 |
| Search for sports content: sports and sports stars | 79.1 | 32.9 | 11.9 | 26.9 | 28.3 | 1.1 | **44.6** |
| Search for information about entertainment: cinema, books, concerts, live shows, and events | 88.5 | 35.5 | 8.5 | 22.9 | 20.2 | 1.5 | **51.6** |
| Search for information related to brands, products, services and companies | 82 | 32.0 | 6.1 | 24.9 | 16.8 | 0.6 | **53.2** |
| Search for information about series, films or TV programmes | 88 | 16.8 | 0.6 | 21.3 | 27.2 | 0.4 | **51.1** |
| Search for information about celebrities | 84.9 | **37.8** | 11.8 | **37.9** | 24.4 | 1.1 | **41.7** |
| Follow professionals in your field of work/study | 73 | 30.8 | 11.6 | 31.0 | 13.0 | 2.4 | **45.6** |
| Search for information about health, diet, nutrition, well-being | 70.8 | 28.1 | 6.8 | 26.5 | 24.9 | 1.2 | **51.5** |
| Search for information about beauty, fashion, style | 73.8 | 28.0 | 4.9 | 34.5 | 28.8 | 1.5 | **44.5** |
| Search for entertaining content | **91.6** | **41.0** | 9.7 | 34.5 | **40.2** | 3.6 | 31.7 |
| Follow your favourite actors, singers, sports stars, and influencers | 83 | **35.4** | 14.9 | **42.3** | 23.4 | 2.5 | 33.0 |
| Play online | 52.7 | 23.4 | 4.1 | 11.2 | 4.9 | 3.3 | **71.1** |
| Buy and sell | 62.1 | 22.3 | 3.6 | 9.8 | 3.5 | 3.3 | **73.6** |
| Publish opinions about social/ political issues | 56.4 | **41.6** | 16.8 | 22.8 | 4.8 | 0.4 | **40.4** |
| Publish criticism/complaints on a public profile | 57.1 | **52.2** | 18.6 | 22.1 | 4.7 | 0.0 | 36.4 |
| Share or recommend sites and links to others | 70.3 | **53.1** | 15.3 | 31.8 | 5.7 | 1.4 | 34.5 |

The highest proportions reached by each activity on every platform appear in bold.

Facebook and Instagram are the two platforms on which most young people said they carried out the following activities: approximately three-quarters posted content of a personal nature on their profile; around two-thirds viewed videos or see photos of friends and family; a slightly lower percentage talked to friends; and about one-third of those who searched for information about celebrities or followed their profiles did so on both platforms

Although Twitter was used by the majority of participants to post personal content (51.7%), to publish criticism or complaints on a public profile (37%), and to give opinions on social or political issues (36%), this was not the platform where most network users claimed to carry out these activities, which mainly occurred on Facebook (52% and 53.1%, respectively), followed by Instagram (22.1% and 22.8%), and Twitter (18.6% and 15.3%).

Once again, Facebook was the preferred social network for talking to family members and sharing or recommending sites and links to others. It shared the leading position with YouTube in searching for entertaining content. In addition, YouTube was chosen more often for watching videos and music. Instagram was used more frequently to follow actors, singers, sports stars, etc.

On the other hand, those who play online, buy and sell, search for information regarding products and services, leisure, health or series on the networks, indicated they used other specific platforms more often for carrying out these activities.

Regarding gender differences (Table 3), a higher percentage of young women indicated they preferred Instagram for activities such as posting personal content on their profile, watching videos, or looking at photos of friends and family, talking to friends, searching for information related to fashion and style, or following celebrities. Whereas young men most frequently used Instagram to search for sports content, buy and sell. On the other hand, more girls showed a preference for YouTube in seeking information about health, diet, nutrition, wellness, beauty, fashion and style, and they used Snapchat to post personal content, watch videos, look at photos of friends and family, and follow celebrities. Young men chose to search for information related to beauty, fashion and style on other social networks to a greater degree than women.

**Table 3.** Use of social networks by platform among young people aged 17 to 24 in the Autonomous Region of Madrid, according to gender. In percentages.

| | Facebook | | Twitter | | Instagram | | YouTube | | Snapchat | | Others | |
|---|---|---|---|---|---|---|---|---|---|---|---|---|
| | **M** | **W** | **M** | **W** | **M** | **W** | **M** | **W** | **M** | **W** | **M** | **W** |
| Publish/upload personal content to your profile | 78.0 | 72.8 | 21.0 | 21.9 | **60.4** | **74.6** [2] | 8.2 | 7.3 | **3.7** | **7.9** [11] | 11.5 | 16.2 |
| Watch videos/see photos of friends and family | 69.0 | 64.7 | 15.6 | 13.9 | **54.3** | **66.1** [3] | 18.0 | 18.3 | **3.2** | **10.2** [12] | 15.9 | 17.6 |
| Talk to friends | 57.5 | 57.2 | 15.1 | 14.0 | **51.8** | **61.3** [4] | 5.9 | 5.8 | 4.6 | 5.2 | 34.3 | 28.7 |
| Talk to family members | 54.5 | 55.0 | 7.6 | 8.7 | **34.4** | **43.6** [5] | 3.7 | 3.7 | 1.2 | 2.6 | 38.9 | 34.5 |
| Search for sports content | 35.9 | 28.4 | 14.0 | 8.7 | **30.7** | **21.3** [6] | 30.5 | 24.9 | 1.1 | 1.0 | 41.3 | 49.7 |
| Search for information about health, diet, nutrition, and well-being | 26.5 | 29.6 | 7.5 | 6.3 | 21.8 | 30.8 | **18.3** | **31.2** [9] | 0.6 | 1.8 | 56.0 | 47.3 |
| Search for information about beauty, fashion and style | 26.7 | 28.8 | 5.1 | 4.8 | **25.8** | **40.2** [7] | **18.5** | **35.7** [10] | 0.8 | 2.0 | **51.5** | **39.9** [14] |
| Follow your favourite actors, singers, sports stars, and influencers | 39.2 | 31.7 | 17.5 | 12.4 | **35.2** | **49.4** [8] | 24.0 | 22.7 | **0.9** | **4.1** [13] | 36.1 | 29.8 |
| Buy and sell | **26.8** | **17.5** [1] | 3.7 | 3.5 | 9.9 | 9.6 | 3.7 | 3.2 | 2.3 | 3.7 | 69.1 | 78.4 |

Only items with statistically significant differences are included for $p < 0.05$, highlighted in bold. [1] $\chi2 = 3.96$, df = 1, $p = 0.047$; [2] $\chi2 = 11.35$, df = 1, $p = 0.001$; [3] $\chi2 = 7.60$, df = 1, $p = 0.006$; [4] $\chi2 = 4.70$, df = 1, $p = 0.03$; [5] $\chi2 = 4.50$, df = 1, $p = 0.034$; [6] $\chi2 = 4.44$, df = 1, $p = 0.035$; [7] $\chi2 = 7.68$, df = 1, $p = 0.006$; [8] $\chi2 = 8.34$, df = 1, $p = 0.004$; [9] $\chi2 = 7.68$, df = 1, $p = 0.006$; [10] $\chi2 = 13.51$, df = 1, $p = 0.001$; [11] $\chi2 = 3.88$, df = 1, $p = 0.049$; [12] $\chi2 = 10.58$, df = 1, $p = 0.001$; [13] $\chi2 = 4.51$, df = 1, $p = 0.034$; [14] $\chi2 = 4.20$, df = 1, $p = 0.023$.

In addition, more young people between 20 and 24 years of age said they chose Facebook to carry out many of the activities compared to the youngest, which is a clear indication of their preference for this network. In contrast, the youngest people chose Twitter to publish personal content. As for Instagram, young people preferred this platform to search for sports information, seek audio-visual content, and follow professionals in the young people's own fields of study. YouTube, however, was chosen most often by the 20–24 year age group to post personal content on their profile. Finally, the youngest people used other networks in higher numbers to talk to friends and family (see Table 4).

**Table 4.** Use of social networks by platform among young people aged 17 to 24 in the Autonomous Region of Madrid, according to age group. In percentages.

| | Facebook | | Twitter | | Instagram | | YouTube | | Snapchat | | Others | |
|---|---|---|---|---|---|---|---|---|---|---|---|---|
| | 17–19 | 20–24 | 17–19 | 20–24 | 17–19 | 20–24 | 17–19 | 20–24 | 17–19 | 20–24 | 17–19 | 20–24 |
| Publish/upload personal content to your profile | **65.7** | **81.2**[1] | **28** | **17.6**[12] | 72.5 | 64.5 | **4.5** | **9.7**[16] | 8.2 | 4.3 | 14.1 | 13.7 |
| Watch videos/see photos of friends and family | **59.3** | **71.4**[2] | 17.6 | 13.0 | 64.9 | 57.2 | 16.3 | 19.2 | 9.3 | 5.1 | 18.5 | 15.6 |
| Talk to friends | **48.4** | **62.8**[3] | 16.1 | 13.7 | 59.1 | 54.9 | 4.4 | 6.7 | 7.2 | 3.5 | **37.3** | **28.0**[17] |
| Talk to family members | **44.5** | **60.9**[4] | 6.7 | 9.0 | 39.1 | 38.8 | 3.3 | 4 | 2.1 | 1.7 | **43.3** | **32.8**[18] |
| Watch videos/music | **33.8** | **46.7**[5] | 10.5 | 8.4 | 44.9 | 37.5 | 66.4 | 63 | 4.1 | 1.7 | 19.1 | 13.7 |
| Search for sports content: sports and sports stars | 34.3 | 30.1 | 10.8 | 12.5 | **33.1** | **23.7**[13] | 32.4 | 26.1 | 0.7 | 1.3 | 41.3 | 46.4 |
| Search for information about entertainment | 29.8 | 38.8 | 7.6 | 9.0 | 25.3 | 21.6 | 21.9 | 19.3 | 0.6 | 2.1 | 52.4 | 51.2 |
| Search for information related to brands, products, services and companies | **26.0** | **35.4**[6] | 6.6 | 5.8 | 30.3 | 21.8 | 17.9 | 16.2 | 0.7 | 0.6 | 53.9 | 52.8 |
| Search for information about series, films, or TV programmes | **23.4** | **33.4**[7] | 8.3 | 6.7 | **26.1** | **18.3**[14] | 28.2 | 26.6 | 0.7 | 0.3 | 51.6 | 50.8 |
| Search for information about celebrities. | 37.5 | 38.1 | 13.9 | 10.6 | 39.4 | 37.1 | 27.1 | 22.8 | 1.5 | 0.8 | 42.9 | 41.0 |
| Follow professionals in your field of work or study | **23.9** | **34.4**[8] | 11.8 | 11.5 | **38.2** | **27.3**[15] | 12 | 13.6 | 3.9 | 1.6 | 47.4 | 44.7 |
| Search for information about health, diet, nutrition, well-being | 24.8 | 30 | 7.6 | 6.4 | 28.5 | 25.3 | 21.9 | 26.6 | 2.3 | 0.6 | 50.5 | 52.1 |
| Search for information about beauty, fashion and style | 24.8 | 29.7 | 5.9 | 4.4 | 37.5 | 32.8 | 28.6 | 29 | 2.1 | 1.2 | 43.1 | 45.4 |
| Search for entertaining content | **34.3** | **45.0**[9] | 10.1 | 9.5 | 35.8 | 33.8 | 43.2 | 38.4 | 4 | 3.3 | 31.6 | 31.7 |
| Follow your favourite actors, singers, sports stars and influencers | 32.3 | 37.3 | 17.5 | 13.5 | 46.9 | 39.7 | 24.6 | 22.6 | 3.9 | 1.7 | 34.5 | 32.0 |
| Play online | 24.9 | 22.5 | 4.5 | 3.8 | 11.9 | 10.7 | 5.8 | 4.4 | 1.9 | 4.2 | 72.2 | 70.4 |
| Buy and sell | 24.4 | 21.3 | 5.5 | 2.6 | 10.8 | 9.2 | 2.7 | 3.8 | 1.0 | 4.0 | 74.2 | 73.2 |
| Publish opinions about social/political issues | **34.3** | **51.8**[10] | 18.9 | 15.8 | 18.9 | 24.7 | 3.3 | 5.6 | 1.1 | 0.0 | 48.2 | 36.6 |
| Publish criticism or complaints on a public profile | 47.0 | 55.2 | 14.6 | 20.9 | 20.9 | 22.7 | 3.8 | 5.2 | 0.0 | 0.0 | 41.5 | 33.4 |
| Share or recommend sites and links to others | **45.1** | **57.6**[11] | 14.5 | 15.8 | 36.3 | 29.3 | 7.1 | 4.9 | 1.6 | 1.2 | 38.7 | 32.2 |

Statistically significant differences for $p < 0.05$ are highlighted in bold. [1] $\chi 2 = 14.75$, df = 1, $p < 0.001$; [2] $\chi 2 = 8.01$, df = 1, $p = 0.005$; [3] $\chi 2 = 10.43$, df = 1, $p = 0.001$; [4] $\chi 2 = 12.20$, df = 1, $p < 0.001$; [5] $\chi 2 = 8.53$, df = 1, $p = 0.003$; [6] $\chi 2 = 3.96$, df = 1, $p = 0.047$; [7] $\chi 2 = 4.82$, df = 1, $p = 0.028$; [8] $\chi 2 = 4.19$, df = 1, $p = 0.041$; [9] $\chi 2 = 5.32$, df = 1, $p = 0.021$; [10] $\chi 2 = 7.87$, df = 1, $p = 0.005$; [11] $\chi 2 = 5.40$, df = 1, $p = 0.020$; [12] $\chi 2 = 7.70$, df = 1, $p = 0.006$; [13] $\chi 2 = 4.35$, df = 1, $p = 0.037$; [14] $\chi 2 = 4.02$, df = 1, $p = 0.045$; [15] $\chi 2 = 4.49$, df = 1, $p = 0.034$; [16] $\chi 2 = 4.67$, df = 1, $p = 0.031$; [17] $\chi 2 = 4.73$, df = 1, $p = 0.030$; [18] $\chi 2 = 5.50$, df = 1, $p = 0.019$.

In conclusion, it is important to note that social class influences the choice of platforms for carrying out some of the activities. It is worth noting that young people from the higher classes chose Facebook more often than those from other socioeconomic levels to publish personal content, search for information related to products, services, and audio-visual content, as well as to look for information about celebrities and follow their profiles. They also used this network to look for data regarding professionals in the young people's own field of work or study, and to find content related to health, beauty, fashion, style, and entertainment. More than other groups, the lower class chose this platform to look for audio-visual information as well as data related to professionals in the young people's own field of work or study. In addition, the latter chose YouTube in greater numbers to watch videos and music, and other platforms for buying and selling.

On the other hand, YouTube had a much greater number of users among young people from the upper-middle and middle class who claimed to have published personal content.

There was a larger presence of the lower-middle class in watching videos and music, and more people from the lower-middle and middle class tended to look for information related to leisure. Those with middle and low social status chose Instagram to post content on their profile. We found that there was a greater preference among the middle classes for other social networking platforms to watch videos and music, search for information about audio-visual entertainment, celebrities, diet, nutrition, wellness, beauty, fashion, and style. The lower class tended to use them mainly for buying and selling (see Table 5).

**Table 5.** Use of social networks by young people from 17 to 24 years of age in the Autonomous Region of Madrid, according to social class. In percentages.

| | | Upper Class | Upper Middle Class | Middle Class | Lower Middle Class | Lower Class |
|---|---|---|---|---|---|---|
| Publish/upload personal content to your profile | Facebook [1] | 83.0 ab | 88.2 b | 71.3 a | 71.8 ab | 64.5 a |
| | Instagram [2] | 53.6 bc | 54.4 b | 75.9 a | 68.1 ab | 70.6 ab |
| | YouTube [3] | 3.5 ab | 12.9 b | 10.5 ab | 2.7 a | 0.0 |
| Watch videos/music | YouTube [4] | 64.3 ab | 51.9 b | 63.5 ab | 72.1 a | 78.0 ab |
| | Others [5] | 5.8 be | 21.9 cde | 21.8 c | 4.4 ab | 13.9 ace |
| Search for sports content | Others [6] | 27.6 b | 47.1 ab | 53.1 a | 38.6 ab | 40.8 ab |
| Search for information about entertainment | YouTube [7] | 8.9 b | 13.4 ab | 23.7 ab | 29.7 a | 3.2 ab |
| Search for information related to brands, . . . | Facebook [8] | 57.6 b | 36.4 ab | 23.4 a | 26.9 a | 39.4 ab |
| Search for information about TV programmes | Facebook [9] | 47.5 b | 34.2 ab | 23.0 a | 25.0 a | 41.5 ab |
| | YouTube [10] | 18.3 a | 16.6 a | 31.3 a | 33.6 a | 24.6 a |
| | Others [11] | 35.0 b | 57.9 ab | 55.3 a | 47.6 ab | 50.6 ab |
| Search for information about celebrities | Facebook [12] | 61.2 b | 44.1 ab | 30.2 a | 31.1 a | 48.9 ab |
| | Others [13] | 19.7 b | 46.7 a | 44.9 a | 45.1 a | 40.7 ab |
| Search for information about health, diet, nutrition, well-being | Facebook [14] | 53.8 b | 23.4 a | 24.3 a | 20.5 a | 34.4 ab |
| | Others [15] | 34.9 b | 65.4 a | 51.4 ab | 53.6 ab | 44.1 ab |
| Search for information about beauty, fashion and style | Facebook [16] | 45.9 b | 22.2 ab | 25.0 a | 23.0 ab | 40.1 ab |
| | Others [17] | 30.8 b | 61.5 a | 44.9 ab | 38.5 ab | 52.0 ab |
| Search for entertaining content | Facebook [18] | 62.7 b | 45.9 ab | 32.8 a | 38.3 a | 48.7 ab |
| Follow your favourite actors, singers, sports stars, and influencers | Facebook [19] | 59.6 b | 41.1 ab | 29.4 a | 24.8 a | 46.1 ab |
| | Others [20] | 20.6 a | 34.1 a | 40.2 a | 25.1 a | 32.7 a |
| Buy and sell | Facebook [21] | 31.2 a | 35.2 a | 17.1 a | 19.6 a | 11.7 a |
| | Others [22] | 55.8 b | 75.2 ab | 77.5 a | 73.9 ab | 88.8 ab |

Only social networking platforms that display significant differences by social class at a significance level of $p < 0.05$ have been included. Note: [1] $\chi2 = 13.17$, df = 4, $p = 0.010$; [2] $\chi2 = 19.12$, df = 4, $p = 0.001$; [3] $\chi2 = 13.68$, df = 4, $p = 0.008$; [4] $\chi2 = 10.93$, df = 4, $p = 0.027$; [5] $\chi2 = 23.49$, df = 4, $p < 0.001$; [6] $\chi2 = 13.22$, df = 4, $p = 0.010$; [7] $\chi2 = 17.96$, df = 4, $p = 0.001$; [8] $\chi2 = 26.66$, df = 4, $p < 0.001$; [9] $\chi2 = 17.38$, df = 4, $p = 0.002$; [10] $\chi2 = 10.53$, df = 4, $p = 0.032$; [11] $\chi2 = 10.40$, df = 4, $p = 0.034$; [12] $\chi2 = 22.57$, df = 4, $p < 0.001$; [13] $\chi2 = 13.61$, df = 4, $p < 0.009$; [14] $\chi2 = 21.02$, df = 4, $p < 0.001$; [15] $\chi2 = 11.86$, df = 4, $p = 0.018$; [16] $\chi2 = 12.99$, df = 4, $p = 0.011$; [17] $\chi2 = 12.72$, df = 4, $p = 0.013$; [18] $\chi2 = 20.48$, df = 4, $p < 0.001$; [19] $\chi2 = 24.58$, df = 4, $p < 0.001$; [20] $\chi2 = 11.02$, df = 4, $p = 0.026$; [21] $\chi2 = 11.48$, df = 4, $p = 0.022$; [22] $\chi2 = 11.58$, df = 4, $p = 0.021$. Values in the same row that do not share the same sub-index (a,b,c) are significantly different at $p < 0.05$ in the bilateral equality test for column proportions.

## 4. Discussion

In general, young people in the Autonomous Region of Madrid appear to be highly intense users of social networks, with 84% accessing these platforms every day, and of these people, many remain connected continuously. The age and gender of the respondents seem to influence their frequency of use, while social class makes almost no difference. Women and younger people (17–19 years of age) were the ones who used social networks on a more continuous basis. This data contrasts with studies carried out by specialized research companies in Spain, which indicated that men were the most intense users. However, it should be noted that the comparison in that study was made with the general population, not just with young people (IAB 2019).

In line with previous studies (Alhabash and Ma 2017; Oh and Syn 2015; Stanley 2015), young people in Madrid showed themselves to be multiplatform users, with a quarter of them accessing all of the social networks mentioned: Facebook, Instagram, YouTube, Twitter and Snapchat, with the first three being the most heavily used sites. This contrasts with other countries such as the USA or Taiwan, where Snapchat was in second place among young people after Instagram (Knight-McCord et al. 2016; Huang and Su 2018),

although it is in line with the data from specialized Spanish research companies (IAB 2019). Thus, Instagram was the network where users spent the greatest amount of time in the context studied, with nearly 60% of its users being connected continuously. The use of YouTube also produced a greater tendency to be constantly connected.

On the other hand, a relationship between the frequency of social network use and the preference for different platforms has been observed. Those who said they continuously use social networks tended to be more active on Instagram and YouTube compared to those who did not engage in continuous use. In large part, these data are consistent with the work of Alhabash and Ma (2017), in which they pointed out that Instagram was the platform that stood out for its intensity of use.

Variables related to age, gender, and social class indicated some differences in the use of platforms, as reflected in the study, which mirrors the study carried out by Blank and Lutz (2017). Unlike these authors, who found no differences in Instagram use with regard to socio-demographic aspects, this platform was used more often by women and the younger group (17–19 years of age) in our context. Women used Snapchat largely, which is in line with the study by Vaterlaus et al. (2016). They especially used it to maintain contact in their most intimate and personal relationships, although it should be noted that young women preferred Instagram for this purpose. Regarding social class, the upper-middle and upper classes made more diverse and varied use in terms of activities on Facebook and Twitter, which is in agreement with a study carried out in Chile (Correa 2016). Especially noteworthy is the preference of the upper class for Facebook in searching for different types of information, as well as information related to products and services, while a very high percentage of young people from the lower class chose other platforms to buy and sell.

Young people also make varied use of social media, as shown in previous studies (Alhabash and Ma 2017; Alrajehi 2016; Kaya and Bicen 2016; Oh and Syn 2015). While Alhabash and Ma (2017) found greater diversity in the reasons why young college students from Michigan use Instagram and Snapchat, this same diversity of motivation among young people in Madrid was found to be related to using Facebook and Instagram, with very similar levels.

Both platforms were used mainly for social and entertainment purposes. Facebook was used to a greater extent to talk to family members and share entertaining content. YouTube was used mainly for the latter purpose, and for watching videos and music. Instagram was largely used by those who followed famous people, such as actors, singers, sports stars, etc.

Even though Twitter stands out for offering a user profile capable of posting personal content, publishing criticism or complaints on a public profile, and communicating opinions on social or political issues (Boczkowski et al. 2018), this is not the platform where most network users carried out these activities. For this purpose, young people reported using Facebook the most, which was also their preferred way to talk to family and friends. This may be in line with the study by Valenzuela et al. (2018), who found that Facebook was perceived to be more effective when protesting on personal networks with stronger links, while Twitter was more oriented toward criticizing people with whom the participants had weaker links. Consequently, this indicates that young people prefer civic participation (defined as a conversation on social and political issues) limited to their personal spaces that are closest to family and friends, which manifests itself via Facebook, as opposed to Twitter, which represents the institutionalized channel of political communication that allows direct contact with people's representatives (parliamentarians, politicians, political parties, interest groups and institutions) (Campos-Domínguez 2017).

On the other hand, those who play online, buy and sell, or seek information related to products and services, leisure, health, or series on the networks, mainly used other specific platforms for these purposes. The findings seem to confirm the conclusions of previous studies (Boczkowski et al. 2018; Fergie et al. 2016), which have suggested that user behavior on different platforms could be influenced by the perception of these platforms and their relevance for certain types of content.

Regarding the way that gender, age and social class are capable of influencing the activities carried out on different platforms, our results show that girls use Instagram in more diverse ways, specifically for social and entertainment purposes, with this being their favorite platform. The older group (20–24 years of age) show greater diversity in using Facebook for social and entertainment purposes as well. Moreover, upper class and upper-middle class people use Facebook more frequently, and in more diverse ways (motivations related to socialization, entertainment, professional information, products, and services) than those of the lower classes. In concordance with Blank and Lutz (2017), this allows us to conclude that sociodemographic characteristics, especially gender, age and social class, play an important role in defining the different ways in which young people use social media, and such influences vary depending on the platform under study, and even on the types of activities being carried out.

On the other hand, the first contribution of this work to the specialised literature is the differentiation of the use of different social networks by young people based on an exhaustive list of activities. In such a way that patterns of consumption and behaviour can be connected with the ability to distinguish consumption from different platforms for different purposes. Secondly, this work has been able to clarify the role played by sociodemographic factors such as gender, age and social class in creating these patterns of use. Thirdly, a comprehensive list of questions on social media actions is presented to the academic community for discussion and potential application on scales for further research.

Among the limitations of this study, we must point out the cross-sectional design of the information gathering process for a specific period of time. This has not allowed us to study changes in the trends of social network use that take place over longer periods of time. Likewise, the use of data exclusively from the Autonomous Region of Madrid might be a limitation that has prevented us from generalizing our results to young people in the rest of the country.

Another aspect to consider relates to the way in which social class has been recorded, using the father's occupation and level of education as a reference, assuming that he was the main income earner in the household; and using the mother as a reference only if he was unemployed or a pensioner. Although this form of measurement may be questionable today, there is still a profound gender inequality in access to the labour market, in income inequalities, and in jobs, which particularly affects women. In fact, 54.7% of Spanish men indicated that they themselves were the person who contributed the most income to the household, while 27.8% of women said the same (CIS 2021).

Furthermore, constant changes in the field of social media result in usage trends that vary from year to year. For future lines of research, we would like to mention the need to carry out longitudinal studies that would allow us to observe the changes in trends and analyze the role of socio-demographic factors in these tendencies.

On the other hand, among the main contributions of this study that bears mentioning is the use of a representative sample from the Autonomous Region of Madrid, as it has allowed us to gain deeper knowledge regarding the patterns of use by young people of the different platforms in this context, and to compare them with other regional and national studies, as well as to clarify the role played by sociodemographic factors such as gender, age and social class in creating these patterns of use.

**Author Contributions:** Conceptualization, A.G.-J., M.C.L.-d.-A., and Y.P.-R.; methodology, M.C.L.-d.-A.; software, M.C.L.-d.-A.; validation, M.C.L.-d.-A., A.G.-J., and Y.P.-R.; formal analysis, M.C.L.-d.-A.; investigation, A.G.-J., M.C.L.-d.-A., and Y.P.-R.; resources, A.G.-J.; data curation, A.G.-J.; writing—original draft preparation, A.G.-J., M.C.L.-d.-A., and Y.P.-R.; writing—review and editing, M.C.L.-d.-A.; visualization, M.C.L.-d.-A. and A.G.-J.; supervision, A.G.-J.; project administration, A.G.-J.; funding acquisition, A.G.-J. All authors have read and agreed to the published version of the manuscript.

**Funding:** This research was funded by Ministry of Economy, Industry and Competitiveness from Spain, under project "Social networks, adolescents and young people: media convergence and digital culture" [CSO2016-74980-C2-2-R].

**Institutional Review Board Statement:** Not applicable.

**Informed Consent Statement:** Not applicable.

**Data Availability Statement:** Not applicable.

**Conflicts of Interest:** The authors declare no conflict of interest.

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
