# Peer review of "Differentiated Uses of Social Networking Platforms among Young People in the Autonomous Region of Madrid"

_socsci, doi:10.3390/socsci10040114_

Round 1
Reviewer 1 Report
The data revealed by the research does not add anything new to the existing data. In the methodology, it is not clear when the interviews were conducted, so in light of the results suggesting that Snapchat is one of the most used networks, the study's data is outdated.
The introduction does not include anything about the pandemic and ignores Tiktok's entry into the market.
A sample of 500 young people in Madrid does not seem to be extrapolated to "young people" in general, from where, Spain, Europe, the world?
This should be duly delimited in the hypotheses, the title and the abstract so as not to mislead.
The bibliography should be updated as this research barely includes studies from 2020 or 2021 and in social networks updating the state of the art is essential.
Author Response
Response to Reviewer 1 Comments
Point 1: The data revealed by the research does not add anything new to the existing data. In the methodology, it is not clear when the interviews were conducted, so in light of the results suggesting that Snapchat is one of the most used networks, the study's data is outdated.
Response 1: The dates for carrying out the fieldwork are set out in the procedure.
Point 2: The introduction does not include anything about the pandemic and ignores Tiktok's entry into the market.
Response 2: No research has been done on Tik Tok. According to the literature, Tik tok has been very popular, especially since the pandemic, and among adolescents (Ballesteros-Herencia 2020; Rivera 2020). We studied young people from 17 to 24 years of age, and the survey is prior to the pandemic. We add some comments in limitations.
Point 3: A sample of 500 young people in Madrid does not seem to be extrapolated to "young people" in general, from where, Spain, Europe, the world?This should be duly delimited in the hypotheses, the title and the abstract so as not to mislead.
Response: This change has been introduced in the title (line 3), included in the abstract (line 17) and in the assumptions (line 125, 131, and 136).
Point 4: The bibliography should be updated as this research barely includes studies from 2020 or 2021 and in social networks updating the state of the art is essential.
Response 4: The literature review is updated and references are included (lines 31-32, 39-41, 53-54, 69-72, 96-100, 114-119, bibliography in: 495-496, 513-515, 528-529, 541-543, 551-555, 559-561).
Best Wishes
Reviewer 2 Report
Interesting paper with still some flaws that need to be addressed.
a) Gender trouble issue: Assumptions like the social class is based on the occupation of the father is no longer acceptable as it is stated in line 138:
occupation and 138 educational level of the father, assuming that he was the one who contributed the most 139 income,
b) Too many Research Questions are made (6), mixing them with methodology. A clear differenciation is needed to avoid confusion.
c) Data is old (2019). A comparison with a Covid situation is needed since the use has been changed (or not, it needs to be proved).
d) Tik Tok is not present and it makes it difficult to analyze this issue without a specific section for it
e) References are not clear and well stated in alfabetical order
References are needed. We recommend to check Journals like El Profesional de la Información or Comunicación y Sociedad to see more state of the art in this field. Also more information on Facebook is needed in terms of references.
Author Response
Response to Reviewer 2 Comments
Point 1: Gender trouble issue: Assumptions like the social class is based on the occupation of the father is no longer acceptable as it is stated in line 138: occupation and 138 educational level of the father, assuming that he was the one who contributed the most 139 income,
Response 1: Included in limitations of the study with an explanation (lines 456-463).
Point 2: Too many Research Questions are made (6), mixing them with methodology. A clear differenciation is needed to avoid confusion.
Response 2: The research questions are in a separate section. In our opinion, it is enough to differentiate them.
Point 3: Data is old (2019). A comparison with a Covid situation is needed since the use has been changed (or not, it needs to be proved).
Response 3: While usage platforms are changing very rapidly, we understand that overall behavioural patterns persist to a greater extent. Included in limitations of the study with an explanation (lines 464-469)
Point 4: Tik Tok is not present and it makes it difficult to analyze this issue without a specific section for it.
Response 4: No research has been done on Tik Tok. At the time of the research, and even afterwards when it experienced an increase during the pandemic, its use was present among those under 16 years of age (Ballesteros-Herencia 2020; Rivera 2020). Because of this, it was decided not to include this Platform in the study.
Point 5: References are not clear and well stated in alfabetical order
Response 5: French indentation (0.75 points) is incorporated for easy reading. The references are arranged alphabetically with the Word tool (lines 482-569).
Point 6: References are needed. We recommend to check Journals like El Profesional de la Información or Comunicación y Sociedad to see more state of the art in this field. Also more information on Facebook is needed in terms of references.
Response 6: We have included new updated references and references from recommended publications. We have not found references to the concrete activities carried out on each platform by young people in the journal Communication and Society. See line 114-119 and 541-543)
The English writing style has been revised. Improvements have been made to the wording.
The authors would like to thank you for your comments and suggestions , which have helped to improve the article significantly.
Best wishes
Reviewer 3 Report
I appreciate the opportunity to review your manuscript. The primary objective of this study is to learn more about the way in which young people in the Autonomous Region of Madrid use social networks, is clear, but not for what. What is the reason to do that research?
The researc describes a reality but we don´t know what could be the use of knowing this data… To subjet an university activity? To be used by an adversiting company? I think that to connect your conclussion to a social science materia indicanting yje implicacions of your results, may be more interesting than do only a descriptive analisys.
Nevertheless, the importance of conducting an inteseccional (sex, age and socio-demographic) study is clear in your study.
The two age stages are well defined, but choosing 17 years old (minors) it is not clear if there was the permissin from their legal guardians or the presence of them during the interviews. (“The interviews were conducted personally in the homes of the respondents”). 533 different homes open the door and stay 25 m. whit the researchers in 15 days? How many people were family members of the same household that could be interviewed? Were they toguether with the interview? Are questions that could be explained.
Another question that arises is to know if the questionaire (“created by the authors”) was validated by experts or was the result of an adaptation of one previously validated and published, What kind of questions were on the questionnaire? Were closed to choose between one option or another? Were there open questions? And if there were, how was the information collected recordered?
The tables are very interesting. However, nota ll of them present the information by sex. For example, the table 1, make imposible to know how many men and women in each age or class; the table 2 does not show us age and sex; the table 3 does not indicate age… To have the data segregated by age (the two segments selected), sex and social class in all the tables would make easier to understand your conclusions.
Good job sowing us in the discussion the weaknesses of that researh. For that reason, the tittle MUST be modified. It can not be “Differentiated uses of social networking platforms among young people”, but a case study in the Autonomous Region of Madrid.
Lastly, regarding to the civic participation analysis, it seems that the information is inferred from open questions. If this is, a critical analysis of the discourse, more qualitative than quantitative, must be done in order to draw the conclusions made in this sections.
Again, I enjoyed reading your work. I think you’re focused on a very intersting and actual key topic if it is connected it whit a for what.
Author Response
Response to Reviewer 3 Comments
Point 1: The research describes a reality but we don´t know what could be the use of knowing this data… To subject an university activity? To be used by an adversiting company? I think that to connect your conclusion to a social science materia indicanting yje implicacions of your results, may be more interesting than do only a descriptive analisys.
Response 1: They have been included in the discussion (lines 441-448)
Point 2: The two age stages are well defined, but choosing 17 years old (minors) it is not clear if there was the permissin from their legal guardians or the presence of them during the interviews. (“The interviews were conducted personally in the homes of the respondents”). 533 different homes open the door and stay 25 m. whit the researchers in 15 days? How many people were family members of the same household that could be interviewed? Were they toguether with the interview? Are questions that could be explained.
Response 2: Currently, Spanish legislation only requires the informed consent of parents or legal guardians for children under 14 years of age. In any case, the explicit consent of the respondent was requested after explaining the type of data that will be collected in the survey and what it will be used for, as well as informing them of its anonymous nature. The fieldwork was carried out by a research firm, following the criteria required by a probabilistic sample and the requirements of current legislation on research issues (lines 177-179)
Point 3: Another question that arises is to know if the questionaire (“created by the authors”) was validated by experts or was the result of an adaptation of one previously validated and published, What kind of questions were on the questionnaire? Were closed to choose between one option or another? Were there open questions? And if there were, how was the information collected recordered?
Response 3: The items included in our research are based on the review of the state of the art, previous research and focus groups. This has furthermore been indicated in lines (167-168). These questions were closed, and the respondents could choose between different options (lines 170 y 171).
Point 4: The tables are very interesting. However, none of them present the information by sex. For example, the table 1, make imposible to know how many men and women in each age or class; the table 2 does not show us age and sex; the table 3 does not indicate age… To have the data segregated by age (the two segments selected), sex and social class in all the tables would make easier to understand your conclusions.
Response 4: Incorporating the values of all the data by age, sex and social class into the tables would greatly increase their size, and these data does not contribute anything because statistically significant differences are not shown for these variables. Otherwise, if it is chosen to enter the data of the items that show significant differences for these variables, the table would be unstructured. In the text those items that show statistically significant differences for these variables are mentioned.
Point 5. The tittle MUST be modified. It can not be “Differentiated uses of social networking platforms among young people”, but a case study in the Autonomous Region of Madrid.
Response 5: Done, it is incorporated in the title (line 3)
Point 6. Regarding to the civic participation analysis, it seems that the information is inferred from open questions. If this is, a critical analysis of the discourse, more qualitative than quantitative, must be done in order to draw the conclusions made in this sections.
Response 6: The analysis of citizen participation is derived from the survey data with closed-ended questions, and specifically it derived from this paragraph: Although Twitter was used by the majority of participants to post personal content (51.7%), to publish criticism or complaints on a public profile (37%), and to give opinions on social or political issues (36%), this is not the platform where most network users claimed to carry out these activities, which mainly occurs on Facebook (52% and 53.1%, respectively), followed by Instagram (22.1% and 22.8%), and Twitter (18.6% and 15.3%).
The English writing style has been revised. Improvements have been made to the wording.
Please see the attachment
WE THANK THE REVIEWERS FOR THEIR CONTRIBUTIONS, WHICH UNDOUBTEDLY CONTRIBUTE TO THE IMPROVEMENT OF THE QUALITY OF THIS WORK.
Best wishes
Round 2
Reviewer 1 Report
Many thanks to the authors for their efforts, the new version of the manuscript has improved considerably, but the tables should be revised as they are difficult to read, especially tables 4 and 5.
I would also include in the introduction a paragraph justifying the importance of the Community of Madrid as a representative sample of the rest of Spain in order to give greater scientific interest and scope to the research.
Author Response
Response to Reviewer 1 Comments
We would like to thank the reviewer for his/her advice.
Point 1. The tables should be revised as they are difficult to read, especially tables 4 and 5.
Response. Table 5 has been revised and two explanatory note have been changed at the bottom of the table in order to improving the understanding of the data. (lines 353):
- Only social networking platforms that display significant differences by social class at a significance level of p < .05 have been included
- Values in the same row that do not share the same sub-index (a, b) are significantly different at p < .05 in the bilateral equality test for column proportions.
Regarding the tables in general, we have tried different ways of exposing the data (for example: including some data in the text), but this one becomes even more cumbersome. We handle a large amount of variables and data, which we have greatly simplified with tables to make it easier to read and understand the results achieved. It is therefore highly complicated to summarise the tables further without losing information on the results achieved.
Point 2. I would also include in the introduction a paragraph justifying the importance of the Community of Madrid as a representative sample of the rest of Spain in order to give greater scientific interest and scope to the research.
Response. A paragraph justifying the importance of the Community of Madrid has been included, but it is necessary to note that these data are not a representative sample of the resto of Spain (lines 116-120).
Best regards
Reviewer 2 Report
The article has positively improved with those new comments.
Author Response
Response to Reviewer 2 Comments
No comments found.
We would like to thank the reviewer for his/her advice.
Best regards

Round 3
Reviewer 1 Report
Although the main problem with this research is the sample chosen, as it does not bring anything new to scientific research on social networks, the text has improved considerably.